# Stated preferences for new HIV prevention technologies among men who have sex with men in India: A discrete choice experiment

**Michael P. Cameron**[1], **Peter A. Newman**[2]*, **Venkatesan Chakrapani**[3],
**Murali Shunmugam**[3], **Surachet Roungprakhon**[4], **Shruta Rawat**[5], **Dicky Baruah**[5],
**Ruban Nelson**[3], **Suchon Tepjan**[6], **Riccardo Scarpa**[1,7]

1 School of Accounting, Finance and Economics, University of Waikato, Hamilton, New Zealand, 2 Factor-Inwentash Faculty of Social Work, University of Toronto, Ontario, Canada, 3 Centre for Sexuality and Health Research and Policy, Chennai, India, 4 Faculty of Science and Technology, Rajamangala University of Technology, Phra Nakhon, Bangkok, Thailand, 5 The Humsafar Trust, Mumbai, India, 6 VOICES-Thailand Foundation, Chiang Mai, Thailand, 7 Business School, Durham University, Durham, United Kingdom

* p.newman@utoronto.ca

**Data Availability Statement:** Data cannot be shared publicly because they contain sensitive patient information and are owned by the

## Abstract

### Introduction

India has the second largest HIV epidemic in the world. Despite successes in epidemic control at the population level, a concentrated epidemic persists among gay and other men who have sex with men (MSM). However, India lags in implementation of biomedical prevention technologies, such as HIV pre-exposure prophylaxis (PrEP). In order to inform scale-up of new HIV prevention technologies, including those in the development pipeline, we assessed willingness to use oral PrEP, rectal microbicides, and HIV vaccines, and choices among product characteristics, among MSM in two major Indian cities.

### Methods

A cross-sectional survey was conducted with a discrete choice experiment (DCE), an established methodology for quantitively estimating end-user preferences in healthcare. Survey participants were randomly assigned to one of three questionnaire versions, each of which included a DCE for one prevention technology. Participants were recruited using chain-referral sampling by peer outreach workers, beginning with seeds in community-based organizations and public sex environments, in Chennai and Mumbai. DCE data were analyzed using random-parameters (mixed) logit (RPL) models.

### Results

Among participants (n = 600), median age was 25 years, with median monthly income of INR 9,000 (~US$125). Nearly one-third (32%) had completed a college degree and 82% were single/never married. A majority of participants (63%) reported condomless anal sex in the past month. The acceptability of all three products was universally high (≥90%). Across all three products, four attributes were significant predictors of acceptability—with efficacy

Humsafar Trust, with restrictions imposed by the Institutional Review Board of the Humsafar Trust. Requests for access to the anonymized data used in this study may be sent to the Humsafar Trust [contact: info@humsafar.org].

**Funding:** This research was supported, in part, by grants from the Canadian Institutes of Health Research (MOP-102512; THA-118570; PI: PAN) and the Canada Foundation for Innovation. VC was supported in part by the DBT/Wellcome Trust India Alliance Senior Fellowship (IA/CPHS/16/1/502667). The funders had no role in study design, data collection and analysis, decision to publish, or preparation of the manuscript.

**Competing interests:** I have read the journal's policy and the authors of this manuscript have the following competing interests: PAN reports serving as an Academic Editor for PLOS ONE. All other authors reported no conflicts of interest. This does not alter our adherence to PLOS ONE policies on sharing data and materials.

consistently the most important attribute, and in decreasing order of preference, side-effects, dosing schedule, and venue. MSM varied in their preferences for product attributes in relation to their levels of education and income, and engagement in sex work and HIV risk behavior.

## Conclusion

This study provides empirical evidence to facilitate the integration of end users' preferences throughout design, testing, and dissemination phases of HIV prevention technologies. The findings also suggest action points and targets for interventions for diverse subgroups to support the effectiveness of combination HIV prevention among MSM in India.

## Introduction

India had an estimated 2.50 million people living with HIV as of 2022, the third largest HIV epidemic in the world after South Africa and Nigeria [1]. Ongoing initiatives by India's National AIDS Control Program along with community-based interventions have contributed to an overall decline in incident HIV infections—by 37% in the decade from 2010 to 2019—representing a degree of success in controlling the epidemic [2]. Nevertheless, this downward trend in HIV incidence and low aggregate HIV prevalence mask tremendous heterogeneity in a country of over 1.4 billion people.

Epidemiological analyses at regional, state, and local levels have revealed extensive differences in HIV incidence and prevalence, and distinct epidemic trajectories within India [3, 4]. Tamil Nadu in southern India and Maharashtra in western India are among 10 states with the highest HIV prevalence [4]. Importantly, immense population disparities persist in HIV incidence and prevalence; the epidemic is concentrated among men who have sex with men (MSM), as well as transgender women, female sex workers, and people who inject drugs, with HIV prevalence estimated from 7 to 28 times higher than the general adult population [2–4].

A predominant focus on aggregated HIV epidemic trends risks a false complacency about the epidemic and overlooking of the most vulnerable populations. In turn, this portends lower resources devoted to HIV prevention and control, and contributes to waning public awareness of HIV, counter to HIV prevention efforts. Moreover, mimicking the HIV response that may have been appropriate two decades ago with interventions constrained to improving HIV knowledge and promoting condom use is incommensurate with the currently expanded toolkit for HIV prevention and the results of ongoing research on new biomedical prevention technologies. In India, despite high levels of both HIV knowledge/awareness and access to free condoms through National AIDS Control Organization (NACO)-led targeted interventions for MSM, suboptimal rates of consistent condom use (~50–55%) persist [5]. This is exacerbated by ongoing stigma and discrimination, with variations in condom use by type of male partner (e.g., primary vs. paying partner) [5], reflecting the challenges of a technology that must be successfully negotiated and applied in each sexual encounter [6, 7].

Oral pre-exposure prophylaxis (PrEP) has proven highly effective in preventing HIV acquisition, with the potential to be a gamechanger in HIV prevention [8]. Ongoing clinical trials of new prevention technologies in the development pipeline, including rectal microbicides and HIV vaccines, may contribute to significant advances in the HIV response. These interventions may obviate some of the challenges of an HIV response predicated largely on the male condom [6, 7]. Nevertheless, biomedical innovations in HIV prevention do not eliminate an

array of individual, social and structural challenges that have hampered the HIV response for four decades—in India and globally. One crucial inroad into bridging new prevention products and real-world challenges, particularly among vulnerable populations, is to integrate community preferences and perspectives in the stages of product development, rollout, and delivery. Understanding end-user preferences is pivotal to the success of global public health programs [9], including HIV prophylaxis [10, 11].

A number of studies internationally have explored preferences for HIV prevention tools and their product characteristics. In a large systematic review, Beckham et al. [12] indicated a substantial increase in the number of published studies eliciting preferences for HIV prevention tools. A variety of methods for preference elicitation were employed, including conjoint analysis, 'willingness-to-try', discrete choice experiments, and contingent valuation. Of the 84 studies included in their review, nearly 90% were conducted in either sub-Saharan Africa (54%) or North America (36%), with only 10 studies in Asia (six of which were conducted in Thailand). Only 54% of studies included key populations at high risk for HIV (e.g., sex workers, MSM). An earlier review [13] similarly found that most stated preference research on HIV prevention and treatment more generally had been concentrated on populations in North America, Europe, and Africa, with only about one-quarter of research studies overall conducted on key populations such as MSM, female sex workers, and transgender women. In preference studies for HIV prevention tools, key populations at risk, particularly in Asia, have been understudied. Moreover, few studies have considered preferences for multiple HIV prevention tools within the same population.

To that end, we sampled MSM populations at high risk for HIV exposure in two large Indian cities and assessed willingness to use three biomedical HIV prevention tools—oral PrEP, rectal microbicides, and HIV vaccines—and choices among product characteristics, with the goal of developing evidence to inform future implementation.

## Methods

The study protocol was approved by the Research Ethics Board of the University of Toronto, Canada (30607) and the Institutional Review Board of The Humsafar Trust, Mumbai, India (IRB00005331). Participants were provided with written information on the study before deciding to take part and indicated their consent by placing an "X" in a box on the consent form before data collection took place. No personally identifying data were collected. Participation was completely voluntary. Individuals who received an invitation to participate could choose whether or not to visit the community organization to do so. Participants who initiated the questionnaire could also opt out of completion with no penalty; however, since all data were anonymized, their responses could not be removed after survey completion, as explained in the informed consent. All individuals had access to HIV prevention services from the partner CBOs whether or not they chose to participate in the study.

### Study design and context

We conducted a cross-sectional survey with an embedded discrete choice experiment (DCE). DCEs are a now established methodology for quantitively estimating end-user preferences in healthcare [14]. Studies have increasingly applied DCEs to discern preferences for HIV prevention, including new prevention technologies, among vulnerable populations [12, 15]. Based on the economic theory of utility maximization [16, 17], DCEs quantify the strength of individual's trade-off preferences [14] between two or more profiles of a product; each profile comprises a set of product attributes (e.g., efficacy, dosing schedule, duration of protection),

delivery options (e.g., site), and cost, with varying values (e.g., 90% vs. 50% efficacy; CBO vs. private hospital venue).

Oral PrEP has been tested extensively and deemed safe and effective across populations, including MSM [18], though it was not yet approved for use in India at the time of the survey. While subsequently approved in India in 2018 [19], PrEP is still not supported in India's national HIV program [20], which largely precludes its use by most vulnerable populations who cannot afford to pay out-of-pocket. HIV surveillance data indicates that PrEP uptake remains very low in India [21], as is characteristic of other countries in the Asia-Pacific region [22]. The other two products modeled in the DCEs, HIV vaccines and rectal microbicides, remain in the development pipeline [23]. A previous analysis focused solely on choices for PrEP attributes among a study subsample [24]; the present analysis includes and contrasts all 3 product DCEs for the full sample, as well as sub-analyses of participant preferences by socio-economic status, HIV risk behavior, and sex work status.

## Sampling and recruitment

Participants were recruited using chain-referral sampling [25] by trained peer outreach workers from community-based organizations (CBOs) in Chennai, Tamil Nadu and Mumbai, Maharashtra. Inclusion criteria were self-identifying as gay, bisexual or "MSM" (used as an identity by some MSM in Chennai), or other local indigenous identities: *kothi* (feminine gender expression, mostly receptive sexual role), "*double-decker*" or versatile (insertive and receptive sexual roles), *panthi* (masculine gender expression, primarily insertive sexual role) [26]; $\geq$ 18 years of age; and sexually active with a man in the previous month. Peer outreach workers recruited initial participants as seeds from public sex environments (cruising areas) and drop-in centers for HIV prevention and education. Seeds invited additional participants until the pre-determined sample size (n = 600) was reached, sufficient for modelling participant preferences for three products with an efficient DCE experimental design [27]. WHO guidelines for DCEs indicate a minimum sample size of 30 is needed per identified subgroup [28]. In the present sample, the subgroups of interest were MSM with low/high income, low/high education, condomless anal sex/no condomless anal sex, and engaged/not engaged in sex work.

We employed several quality control measures to mitigate risks in chain-referral sampling; for example, individuals might present at the study site, having not been referred, and may misrepresent their sexual identity or behavior to enable them to receive study honoraria. For one, peer recruiters were employed by existing CBOs and well-established in the MSM community and recruitment sites; and they were trained to assess the eligibility of potential participants. Second, each recruiter maintained a log register that tracked individuals referred by seeds—directly invited by the peer recruiter—and then cross-checked whether the same individuals were also referred by other seeds (based on duplication in names, ages, and phone numbers in the logs). Third, pre-numbered coupons were issued to all participants, each with a unique identification number (UID), to invite others in the chain-referral process. Peer recruiters tracked new participants with these coupons, thus indicating the recruiter for each individual. Absent an initial invitation as a seed, or a coupon, other individuals were not able to participate. These measures facilitated the chain-referral process and prevented misrepresentation of one's eligibility and duplicate enrollment.

## DCE experimental design

The DCE design follows widely accepted approaches for DCEs in health applications [14], including methods for selecting the attributes and values modeled [12]. Specifically, the design was developed using a Bayesian D-error minimizing approach with dummy coded variables

obtained using Ngene software (ChoiceMetrics, Sydney, Australia). This Bayesian efficient design allows for the incorporation of information based on empirically informed priors—estimates based on previous data—to select the attributes and values modeled, and to inform model coefficients [29]. Bayesian D-error minimization algorithmically selects a subset of the full factorial design that will provide an efficient estimate (i.e., one with low sampling variance), and ensures the validity and logic of the experimental design [27]. The goal of this method is to create an optimal design for providing accurate assessments of participant preferences [29].

## DCE development

We used a multistage process to develop the DCEs, in accordance with best practice guidelines [12]. First, we conducted a literature review and formative qualitative research on each prevention technology with the focal study population [7, 30, 31]. Based on this information, we identified the most salient attributes for these HIV prevention technologies: concerns about the degree of efficacy, dosing frequency, and side effects emerged across our qualitative research on hypothetical HIV vaccines, microbicides, and PrEP [7, 30, 31]. Additionally, MSM expressed concerns about stigma if it were disclosed that they were accessing these HIV prevention products, such as being judged as sexually "promiscuous" or being "outed" to their families as MSM. This invoked the potential importance of the venue for accessing HIV prevention products as a determinant of product acceptability and use.

Second, we conducted a pilot DCE study with MSM (n = 16) recruited from CBO partners in India; this provided a priori guidance on the model coefficients and informed the selection of attributes (e.g., efficacy) and levels (e.g., 50% vs 99%) for each product in the DCE [32]. It also supported the feasibility of implementing DCEs with this population.

Third, we incorporated methodological recommendations for choice elicitation tasks [32]: alternative attribute levels (e.g., 50% vs 99% efficacy; none vs. minor side effects) need to be sufficiently distinct to be comprehensible to participants, most of whom are unlikely to be able to discern the difference between 50% and 65% efficacy, for example, or a 15% vs. 25% chance of fever or headaches. Our recruitment aimed to reach vulnerable populations of largely low socioeconomic status MSM, many without college education and with limited economic opportunities, a substantial proportion of whom rely on sex work for income. This underscored the importance of selecting attribute levels to facilitate comprehension among participants with low numeracy.

Finally, the very high efficacy of oral PrEP in clinical trials informed our use of 99% as a level. As a result, 99% efficacy was used as the upper value for all 3 products, in contrast with 50% to clearly signify partial efficacy.

Based on our formative research and pilot DCE study, we constructed product alternatives, each comprised of five dichotomous attributes. Pictorial representations depicting each attribute, also tested in formative research, were used in the DCEs (see Fig 1).

Levels of efficacy (99% or 50%) and side effects (none or minor) were invariant across the three products. To facilitate understanding, participants were instructed that "99% effective" means it would, on average, protect 99 out of 100 people exposed to HIV, as depicted graphically in cards presented on a tablet screen (Fig 1). For dosing, we identified salient attributes and levels for each product, as revealed in our formative research, and clinical guidelines in the case of oral PrEP: PrEP—4 times/week vs. daily; microbicide—no applicator vs. applicator; and HIV vaccine—1 dose/year vs. 3 doses/year. Cost was expressed in Indian rupees (INR), and varied between products, from per-use for microbicides, to monthly for PrEP, to yearly

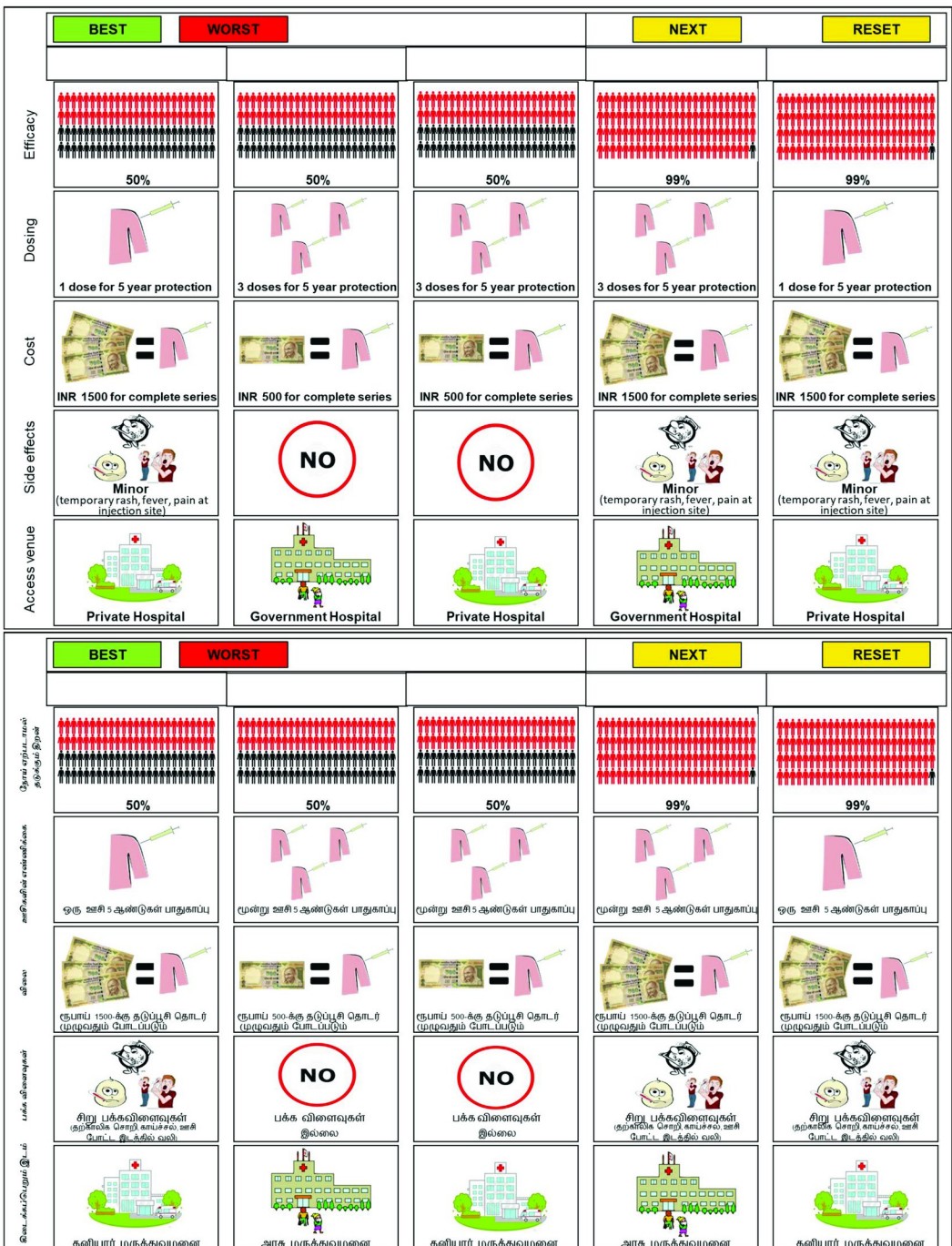

**Fig 1. Sample choice scenario for HIV vaccines presented on the tablet screen for the DCE, in English and Tamil.**

for a vaccine, reflecting the dosing schedules of each product. Venue was modelled as private hospital vs. government hospital for PrEP and HIV vaccines, given the need for administration by a medical professional; for rectal microbicides, we used pharmacy vs. CBO, anticipating potential distribution outside of formal medical settings, similar to condoms.

## Data collection

The survey was programmed on tablet devices in Tamil and Hindi language and self-administered using Tablet-Assisted Self Interviewing (TASI). From 1 December 2016 to 31 March 2017, eligible participants completed the survey at a local CBO office in each city. Participants received an honorarium of INR 300 (~US $6) for completing the one-time survey. Participants also received INR 50 (~US $1) for each successful referral of other MSM to the study.

To mitigate potential respondent fatigue, we created three versions of the questionnaire, each of which was identical except for the DCE. Participants were randomly assigned to one of the three versions including a DCE for one prevention technology (i.e., oral PrEP, microbicide, or vaccine). The DCE experimental design (based on Bayesian D-error minimization) contained 32 choice scenarios of five hypothetical product alternatives for each of the three products. To reduce cognitive burden on participants, these were each blocked in four groups of eight choice scenarios. Random allocation to survey versions and blocking within each product DCE were programmed on the Tablet devices.

Each participant was presented with a description of the product in the survey version to which they were randomized (see S1 Appendix), followed by a series of eight choice scenarios. Each scenario consisted of five product cards displayed on the tablet screen, with each card having a combination of five attributes. Participants were instructed to select the "best" and "worst" options for themselves out of five, and to drag and drop the "best" label into a box on top of their most preferred product card and the "worst" label into a box on top of their least preferred product card on the tablet screen (Fig 1). Participants then repeated the best–worst task with the remaining three cards, with the final (5th) card assigned by default to third place. Participants first conducted a practice choice task on the tablet to familiarize themselves with the method, after which this double best–worst procedure was repeated eight times.

The design of the instrument follows WHO guidelines for DCEs [28], using methods we have previously implemented with MSM in Thailand [32]. Previous stated preference research supports participants' capacity to rate eight choice scenarios, with respondent fatigue occurring well beyond this threshold [33, 34]. Moreover, in earlier feasibility studies of using stated preference methods for HIV prevention technologies with eight choice scenarios among several key populations at risk for HIV, the majority of participants indicated the task was easy/somewhat easy to complete, with some describing the choice scenarios as more engaging than a series of individual Likert-type items [35].

## Measures

Acceptability of each product was measured by the response to the question, "How likely would you be to [take PrEP/use a rectal microbicide/get an HIV vaccine] if it was prescribed by a medical doctor?", measured on a four-point Likert scale (very unlikely; unlikely; likely; very likely; don't know). Responses were coded as 1 if the participant responded that they were very likely to accept the product, and 0 otherwise. Differences in acceptability between different subgroups of the sample were assessed using Chi-squared tests.

The DCE data were analyzed using random-parameters (mixed) logit (RPL) models with the assumption of fully correlated random normal coefficients (i.e., correlations between the four independent variables). Due to differences in the attributes between the three products, and potential differences in research participant interpretations of the levels of the attributes in common across the products (efficacy and side effects), DCE analyses were run separately for each product.

All main effects were included as random normal parameters except for cost, which was fixed. This enables estimates of the distribution of marginal willingness-to-pay (mWTP) to be

estimated from the model [16]. Marginal willingness-to-pay represents the relative desirability of each product attribute, explicitly measured in monetary terms [36]. It can be estimated as the ratio of the coefficient of the non-monetary random attribute to the coefficient on the cost attribute [37].

Additional robustness checks using two alternative model specifications (random-parameters logit with independent random normal coefficients; and conditional logit) yielded qualitatively similar results. Hence, in this paper we present only results from the correlated RPL models. Our preference for the correlated RPL models is supported by the statistical significance of all random parameters in the main analysis, and the smaller Akaike's Information Criterion and Bayesian Information Criterion for these models.

Based on an initial analysis of the sub-group differences in product acceptability, we also conducted analyses of the DCE data, stratified by: (1) income (differentiating between those with monthly income above or below INR10,000 per month); (2) education level (differentiating between those with a college degree, and those with less than college education); (3) whether the research participant had reported any condomless anal sex in the past month; and (4) whether the research participant was engaged in sex work. All stratified analyses were conducted independently, using correlated RPL models, as for the full sample. All analyses were conducted in Stata 16.1 (College Station, Tex., USA).

## Results

Of the 600 research participants, seven were excluded from the analysis due to self-disclosed HIV-positive status. Table 1 presents summary statistics for the remaining sample ($n = 593$).

**Table 1. Sample summary statistics.**

| Variable | Total (%) | PrEP | Microbicide | Vaccine | $X^2$ | $p$-value |
|---|---|---|---|---|---|---|
| **Full analytic sample** | 593 (100.0) | 197 (100.0) | 197 (100.0) | 199 (100.0) | | |
| **Age group** | | | | | | |
| ≤ 25 years | 317 (53.5) | 105 (53.3) | 105 (53.3) | 107 (53.8) | 0.01 | 0.994 |
| > 25 years | 276 (46.5) | 92 (46.7) | 92 (46.7) | 92 (46.2) | | |
| **Monthly income**[a] | | | | | | |
| <INR10,000 (US$138) | 283 (51.3) | 89 (48.1) | 97 (54.5) | 97 (51.3) | 1.48 | 0.477 |
| ≥INR10,000 (US$138) | 269 (48.7) | 96 (51.9) | 81 (45.5) | 92 (48.7) | | |
| **Education** | | | | | | |
| Higher secondary school or lower | 402 (67.8) | 131 (66.5) | 142 (72.1) | 129 (64.8) | 2.61 | 0.271 |
| College degree or higher | 191 (32.2) | 66 (33.5) | 55 (27.9) | 70 (35.2) | | |
| **Marital status** | | | | | | |
| Never married | 484 (81.6) | 160 (81.2) | 157 (79.7) | 167 (83.9) | 1.21 | 0.546 |
| Married, Separated, Divorced or Widowed | 109 (18.4) | 37 (18.8) | 40 (20.3) | 32 (16.1) | | |
| **Sexual or sexual role-based identity** | | | | | | |
| Kothi/double-decker/gay | 408 (68.8) | 134 (68.0) | 142 (72.1) | 132 (66.3) | 1.61 | 0.447 |
| Other | 185 (31.2) | 63 (32.0) | 55 (27.9) | 67 (33.7) | | |
| **Condomless anal sex (last month)** | | | | | | |
| Yes | 374 (63.1) | 115 (58.4) | 124 (62.9) | 135 (67.8) | 3.81 | 0.149 |
| No | 219 (36.9) | 82 (41.6) | 73 (37.0) | 64 (32.2) | | |
| **Sex work** | | | | | | |
| No | 346 (58.4) | 109 (55.3) | 116 (58.9) | 121 (60.8) | 1.26 | 0.534 |
| Yes | 247 (41.7) | 88 (44.7) | 81 (41.1) | 78 (39.2) | | |

[a] Monthly income was not reported for 41 participants

INR, Indian rupee

Overall, the median age of participants was 25 years (mean 26.5, SD 6.4) and median monthly income was INR 9,000 (mean INR 10,191, SD 8,597) or US$125 (1 US$ = INR 72). Nearly one third (32.2%) had completed a college degree and 81.6% had never been married to a woman. Participants reported a variety of sexual or sexual role-based identities, including kothi (29.3%), versatile/double-decker (23.6%), gay (15.9%), bisexual (17.5%) and panthi (13.5%). A majority of participants (63.1%) reported any condomless anal sex in the past month. Similar numbers of participants were allocated to the PrEP, rectal microbicide, and vaccine evaluation tasks, and there were no statistically significant differences between the proportions of each group allocated to each task (Table 1). This demonstrates that the randomization to product evaluation task (questionnaire/DCE type) was effective.

The acceptability of all three products was universally high: 93.9% of participants self-reported to be likely or very likely to take oral PrEP (43.2% likely; 50.8% very likely) if it was prescribed by a medical doctor. For rectal microbicides and an HIV vaccine, the corresponding proportions were 94.4% (47.7% likely; 46.7% very likely) and 90.0% (43.2% likely; 46.7% very likely). As shown in Table 2, there were few statistically significant differences in the proportions of participants from different groups being 'very likely' to accept the products. Higher income participants were significantly less likely than lower income participants to report being very likely to take PrEP. In contrast, when compared to less educated participants, a higher proportion of college educated participants reported being very likely to take a vaccine.

The results of the main DCE analysis are reported in Table 3. The estimates are reported as odds ratios (exponentiated coefficients) and marginal willingness-to-pay (mWTP), each with 95% confidence intervals. Odds ratios and mWTP cannot generally be compared across the different products because of differences in definitions of the attributes, meaning that mWTP refers to different dosage units. The exceptions are cost and efficacy. Within products, efficacy is consistently the most important attribute. Participants had nearly 20 times higher log odds of selecting PrEP with high efficacy (99%) than PrEP with low efficacy (50%), and more than 10 times higher log odds of selecting microbicides and a vaccine with high efficacy than one with low efficacy. In terms of mWTP, this translates into participants being willing to pay INR 7,259 (~US$101) more for high-efficacy PrEP than for low-efficacy PrEP, INR 78 (~US$1) more for a high-efficacy microbicide than for a low-efficacy microbicide, and INR 5,523 (~US$77) more for a high-efficacy vaccine than for a low-efficacy vaccine. The substantially lower mWTP for the microbicide reflects that microbicides were presented to research participants per dose, whereas PrEP was presented as 30 pills, and vaccines as the cost for one year of protection. This also explains why research participants were less sensitive to the cost differences for microbicides than for the other two products (as demonstrated by the odds ratio being much closer to one for microbicides than for the other two products).

The absence of minor side effects was the second most important attribute for all three products, with mWTP between 37% (PrEP) and 66% (vaccine) as large as for efficacy. Dosing was the third most important attribute for all three products, while the venue for receiving the product was consistently rated the least important attribute, and was not a statistically significant determinant of product choice in the case of vaccines.

Table 4 presents the DCE analysis, stratified by income (differentiating between those with monthly income above or below INR10,000 per month). Low-income participants were less sensitive to the price of PrEP than high-income participants. This is shown in the difference in preferences for PrEP attributes, with low-income participants having a greater preference for high efficacy than high-income participants (Odds Ratio [OR], 24.95 vs. 19.06), and higher mWTP (INR 7,651 vs. INR 5,874). This may also reflect the significantly higher product acceptability for PrEP among low-income participants reported in Table 2. Preferences for other PrEP attributes were similar between the two income groups. For vaccines, similar to

**Table 2. Product acceptability.**

| Variable | Number "very likely" to use product (%) | | |
|---|---|---|---|
| | PrEP (n = 197) | Microbicide (n = 197) | Vaccine (n = 199) |
| **Full Sample** | 100 (50.8) | 92 (46.7) | 93 (46.7) |
| **Age group** | | | |
| ≤ 25 years | 55/105 (52.4) | 52/105 (49.5) | 50/107 (46.7) |
| > 25 years | 45/92 (48.9) | 40/92 (43.5) | 43/92 (46.7) |
| $X^2$ (p-value) | 0.24 (0.627) | 0.72 (0.396) | <0.01 (0.999) |
| **Monthly income** | | | |
| < INR10,000 (US$138) | 53/89 (59.6) | 46/97 (47.4) | 42/97 (43.3) |
| ≥ INR10,000 (US$138) | 42/96 (43.8) | 38/81 (46.9) | 50/92 (54.4) |
| $X^2$ (p-value) | 4.62 (0.032)** | 0.005 (0.946) | 2.31 (0.129) |
| **Education** | | | |
| Higher secondary school or lower | 61/131 (46.6) | 65/142 (45.8) | 52/129 (40.3) |
| College degree or higher | 39/66 (59.1) | 27/55 (49.1) | 41/70 (58.6) |
| $X^2$ (p-value) | 2.76 (0.097) | 0.18 (0.676) | 6.08 (0.014)** |
| **Marital status** | | | |
| Not married | 80/160 (50.0) | 74/157 (47.1) | 75/167 (44.9) |
| Married, Separated, Divorced or Widowed | 20/37 (54.1) | 18/40 (45.0) | 18/32 (56.6) |
| $X^2$ (p-value) | 0.20 (0.657) | 0.06 (0.809) | 1.39 (0.239) |
| **Sexual or sexual role-based identity** | | | |
| Kothi/double-decker/gay | 70/134 (52.2) | 70/142 (49.3) | 64/132 (48.5) |
| Other | 30/63 (47.6) | 22/55 (40.0) | 29/67 (43.3) |
| $X^2$ (p-value) | 0.37 (0.545) | 1.38 (0.241) | 0.48 (0.487) |
| **Condomless anal sex (last month)** | | | |
| Yes | 56/115 (48.7) | 59/124 (47.6) | 60/135 (44.4) |
| No | 44/82 (53.7) | 33/73 (45.2) | 33/64 (51.6) |
| $X^2$ (p-value) | 0.47 (0.492) | 0.10 (0.747) | 0.88 (0.347) |
| **Sex work** | | | |
| No | 58/109 (53.2) | 55/116 (47.4) | 56/121 (46.3) |
| Yes | 42/88 (47.7) | 37/81 (45.7) | 37/78 (47.4) |
| $X^2$ (p-value) | 0.59 (0.444) | 0.06 (0.810) | 0.03 (0.873) |

* $p<0.1$;

** $p<0.05$

INR, Indian rupee

PrEP, low-income participants had a greater preference for high efficacy than high-income participants (OR, 13.48 vs. 10.40), and greater mWTP (INR 5,539 vs. INR 4,852). In contrast, for microbicides, the high-income group had a greater preference and higher mWTP for high efficacy than the low-income group (with preference for other attributes being similar between the two groups). Finally, vaccines are the only product where a significant preference for venue emerges, with low-income participants significantly preferring that vaccines be available through government hospitals rather than private hospitals.

Table 5 presents the DCE analysis, stratified by education (differentiating between those with a college degree, and those with less than college education). For PrEP, low-education participants had both a higher preference for high efficacy than high-education participants (OR, 21.98 vs. 16.94) and high mWTP (INR 7,610 vs. INR 6,689). High-education participants showed moderately greater preference for avoiding side effects, and for less frequent dosing,

**Table 3. Main DCE analysis.**

| Product/Attribute (Levels) | Main model—PrEP | | Main model—Microbicide | | Main model—Vaccine | |
|---|---|---|---|---|---|---|
| | Adjusted odds ratio (95% CI) | Marginal willingness-to-pay (95% CI) | Adjusted odds ratio (95% CI) | Marginal willingness-to-pay (95% CI) | Adjusted odds ratio (95% CI) | Marginal willingness-to-pay (95% CI) |
| **Cost** (INR 000s for PrEP, Vaccine; INR for microbicide) | 0.662*** [0.605,0.724] | | 0.970*** [0.966,0.974] | | 0.648*** [0.598,0.703] | |
| **Side effects** (1 = minor, vs. none) | 0.294*** [0.232,0.372] | -2698 [-3434,-2584] | 0.373*** [0.341,0.409] | -33 [-37,-29] | 0.206*** [0.159,0.266] | -3648 [-4604,-2916] |
| **Efficacy** (1 = 99%, vs. 50%) | 19.978*** [13.707,29.119] | 7259 [5662,9639] | 10.595*** [8.16,13.757] | 78 [70,87] | 10.946*** [8.003,14.971] | 5523 [4812,6434] |
| **Dosing** (1 = 4 times/week, vs. daily for PrEP; no applicator vs. applicator for microbicide; 1 dose/year vs. 3 doses/year for vaccine) | 2.023*** [1.81,2.26] | 1707 [1422,2123] | 0.776*** [0.683,0.881] | -8 [-12,-4] | 0.548*** [0.49,0.614] | -1387 [-1772,-1076] |
| **Venue** (1 = private hospital, vs. government hospital for PrEP, Vaccine; pharmacy vs. CBO for microbicide) | 0.907*** [0.845,0.975] | -236 [-406,-60] | 0.879*** [0.796,0.969] | -4 [-8,-1] | 0.919 [0.821,1.03] | -194 [-469,72] |

* *p*<0.1;

** *p*<0.05;

*** p<0.01

INR, Indian rupee

than low-education participants. These differences were not as apparent between low-income and high-income groups (Table 4).

Only low-education participants had a statistically significant preference for venue, preferring to receive PrEP from a government hospital rather than a private hospital. This difference was not apparent between low-income and high-income groups. For vaccines, the greater preference and higher mWTP for high efficacy among low-education participants, and greater preference and higher mWTP for avoiding side effects among high-education participants were also apparent. However, there was little difference in preference for dosing between groups, although there was higher mWTP for annual dosing among high-education participants than among low-education participants (INR 2,076 vs. INR 1,214). This seeming incongruence between preferences and mWTP arises because low-education participants showed a much greater sensitivity to the price of vaccines than high-education participants (odds ratio 0.61 vs. 0.74). This difference in price sensitivity was not apparent for either of the other two products. Finally, similar to the results stratified by income, the high-education group showed greater preference for, and higher mWTP for, efficacy of microbicides. For this product, venue was statistically significant only for low-education participants, who preferred to receive microbicides through a CBO rather than a pharmacy.

Table 6 indicates how product attribute preferences differ by risk behavior, by stratifying the DCE analysis by an indicator of whether the participant had engaged in condomless anal sex within the previous month. Similar to the income- and education-stratified results presented earlier, for PrEP it is the higher-risk group that demonstrates a greater preference (OR, 23.57 vs. 13.34) and higher mWTP for efficacy (INR 7,170 vs. INR 6,997). The high-risk group is also somewhat more sensitive to the price of PrEP (odds ratio 0.64 vs. 0.69), which may also reflect that they are a lower-income group. In terms of dosing, the low-risk group displays a greater preference (OR 2.14 vs. 1.94) and greater mWTP (INR 2,048 vs. INR 1,503) for less frequent dosing of PrEP.

**Table 4. DCE analysis, stratified by income.**

| Product/Attribute (Levels) | Low income (<10,000 INR/month) | | High income (>10,000 INR/month) | |
|---|---|---|---|---|
| | Adjusted odds ratio (95% CI) | Marginal willingness-to-pay (95% CI) | Adjusted odds ratio (95% CI) | Marginal willingness-to-pay (95% CI) |
| **PrEP** | | | | |
| Cost (INR 000s) | 0.657*** [0.570,0.757] | | 0.606*** [0.52,0.705] | |
| Side effects (1 = minor, vs. none) | 0.266*** [0.177,0.400] | -3149 [-4056,-2484] | 0.244*** [0.167,0.356] | -2814 [-3389,-2381] |
| Efficacy (1 = 99%, vs. 50%) | 24.954*** [15.045,41.388] | 7651 [5500,11759] | 19.057*** [11.687,31.076] | 5874 [4183,8864] |
| Dosing (1 = 4 times/week, vs. daily) | 2.241*** [1.795,2.798] | 1919 [1400,2803] | 2.089*** [1.787,2.441] | 1468 [1156,1979] |
| Venue (1 = private hospital, vs. government hospital) | 0.939 [0.822,1.071] | -151 [-454,189] | 0.937 [0.845,1.04] | -129 [-330,88] |
| **Microbicide** | | | | |
| Cost (INR) | 0.969*** [0.961,0.976] | | 0.969*** [0.963,0.975] | |
| Side effects (1 = minor, vs. none) | 0.338*** [0.289,0.397] | -34 [-43,-28] | 0.350*** [0.313,0.391] | -33 [-41,-28] |
| Efficacy (1 = 99%, vs. 50%) | 11.954*** [7.222,19.786] | 78 [65,95] | 14.789*** [9.630,22.711] | 85 [74,101] |
| Dosing (1 = no applicator vs. applicator) | 0.745** [0.585,0.948] | -9 [-17,-2] | 0.808** [0.676,0.965] | -7 [-12,-1] |
| Venue (1 = pharmacy vs. CBO) | 0.878 [0.724,1.066] | -4 [-12,2] | 0.949 [0.820,1.099] | -2 [-7,3] |
| **Vaccine** | | | | |
| Cost (INR 000s) | 0.625*** [0.549,0.712] | | 0.617*** [0.544,0.701] | |
| Side effects (1 = minor, vs. none) | 0.164*** [0.099,0.273] | -3850 [-5488,-2684] | 0.176*** [0.111,0.279] | -3595 [-5156,-2495] |
| Efficacy (1 = 99%, vs. 50%) | 13.477*** [8.461,21.466] | 5539 [4508,7145] | 10.401*** [6.941,15.587] | 4852 [4080,6007] |
| Dosing (1 = 1 dose/year vs. 3 doses/year) | 0.509*** [0.419,0.619] | -1437 [-2167,-938] | 0.537*** [0.457,0.631] | -1289 [-1855,-888] |
| Venue (1 = private hospital, vs. government hospital) | 0.810** [0.679,0.967] | -448 [-918,-59] | 0.923 [0.771,1.105] | -166 [-582,211] |

\* p<0.1;

\*\* p<0.05;

\*\*\* p<0.01

INR, Indian rupee

The results are similar for vaccines, with the higher-risk group showing greater preference for efficacy (OR, 11.41 vs. 9.59). However, in terms of mWTP for vaccines, the results are reversed with the lower-risk group having higher mWTP (INR 6,398 vs. INR 5,156). This reflects the much greater price sensitivity among the high-risk group (OR on cost 0.62 vs. 0.70), again probably reflecting that higher risk intersects with lower income status. Unlike PrEP, there is a strong difference in preferences to avoid side effects of vaccines, with the high-risk group showing stronger preferences for (OR, 0.17 vs. 0.33) and higher mWTP (INR 3,758 vs. INR 3,150) to avoid minor side effects.

Unlike the analyses stratified by income or education, the results for microbicides show similar results to the other products when stratified by risk behavior. High-risk participants

**Table 5. DCE analysis, stratified by education.**

| Product/Attribute (Levels) | Low education (higher secondary school or lower) | | High education (college degree or higher) | |
|---|---|---|---|---|
| | Adjusted odds ratio (95% CI) | Marginal willingness-to-pay (95% CI) | Adjusted odds ratio (95% CI) | Marginal willingness-to-pay (95% CI) |
| **PrEP** | | | | |
| Cost (INR 000s) | 0.666*** [0.600,0.739] | | 0.655*** [0.552,0.777] | |
| Side effects (1 = minor, vs. none) | 0.313*** [0.237,0.413] | -2864 [-3462,-2380] | 0.262*** [0.171,0.400] | -3167 [-4323,-2483] |
| Efficacy (1 = 99%, vs. 50%) | 21.978*** [14.698,32.862] | 7610 [5897,10407] | 16.944*** [10.450,27.474] | 6689 [4700,11115] |
| Dosing (1 = 4 times/week, vs. daily) | 1.935*** [1.675,2.235] | 1626 [1263,2156] | 2.217*** [1.836,2.678] | 1882 [1378,2920] |
| Venue (1 = private hospital, vs. government hospital) | 0.878*** [0.803,0.961] | -319 [-537,-99] | 0.980 [0.867,1.108] | -48 [-328,285] |
| **Microbicide** | | | | |
| Cost (INR) | 0.971*** [0.967,0.976] | | 0.968*** [0.960,0.976] | |
| Side effects (1 = minor, vs. none) | 0.369*** [0.333,0.409] | -34 [-40,-29] | 0.380*** [0.319,0.452] | -30 [-38,-24] |
| Efficacy (1 = 99%, vs. 50%) | 9.772*** [7.313,13.060] | 78 [69,89] | 12.819*** [6.925,23.730] | 78 [64,95] |
| Dosing (1 = no applicator vs. applicator) | 0.767*** [0.661,0.89] | -9 [-14,-4] | 0.777** [0.620,0.972] | -8 [-14,-1] |
| Venue (1 = pharmacy vs. CBO) | 0.837*** [0.751,0.934] | -6 [-11,-2] | 0.976 [0.786,1.213] | -1 [-9,5] |
| **Vaccine** | | | | |
| Cost (INR 000s) | 0.606*** [0.551,0.668] | | 0.737*** [0.639,0.850] | |
| Side effects (1 = minor, vs. none) | 0.192*** [0.140,0.262] | -3304 [-4250,-2561] | 0.248*** [0.168,0.365] | -4568 [-8622,-2848] |
| Efficacy (1 = 99%, vs. 50%) | 13.668*** [9.567,19.528] | 5229 [4573,6078] | 6.999*** [4.273,11.462] | 6374 [4764,10260] |
| Dosing (1 = 1 dose/year vs. 3 doses/year) | 0.545*** [0.480,0.619] | -1214 [-1586,-916] | 0.531*** [0.452,0.624] | -2076 [-3793,-1359] |
| Venue (1 = private hospital, vs. government hospital) | 0.920 [0.803,1.054] | -167 [-460,110] | 0.907 [0.761,1.082] | -318 [-1007,305] |

\* p<0.1;

\*\* p<0.05;

\*\*\* p<0.01

INR, Indian rupee

showed greater preference and mWTP for efficacy than low-risk participants. Both groups showed relatively similar results for the other attributes of microbicides.

Another indicator of HIV risk is engagement in sex work. Table 7 presents the DCE analysis, stratified by whether participants were engaged in sex work or not. For PrEP, participants engaged in sex work were slightly less price sensitive than those not engaged in sex work (OR for cost 0.69 vs. 0.64). Participants engaged in sex work also demonstrated a greater preference (OR 24.29 vs. 15.71) and higher mWTP (INR 8,675 vs. INR 6,096) for higher efficacy PrEP than those not engaged in sex work. However, those not engaged in sex work had a greater preference for receiving PrEP from a government hospital, while those engaged in sex work had no statistically significant preference over venue.

**Table 6. DCE analysis, stratified by condomless anal sex.**

| Product/Attribute (Levels) | Condomless anal sex in the previous month | | No condomless anal sex in the previous month | |
|---|---|---|---|---|
| | Adjusted odds ratio (95% CI) | Marginal willingness-to-pay (95% CI) | Adjusted odds ratio (95% CI) | Marginal willingness-to-pay (95% CI) |
| **PrEP** | | | | |
| Cost (INR 000s) | 0.644*** [0.571,0.725] | | 0.691*** [0.602,0.792] | |
| Side effects (1 = minor, vs. none) | 0.294*** [0.216,0.399] | -2779 [-3308,-2353] | 0.298*** [0.211,0.422] | -3266 [-4505,-2514] |
| Efficacy (1 = 99%, vs. 50%) | 23.573*** [14.888,37.325] | 7170 [5399,10083] | 13.335*** [8.002,22.221] | 6997 [4982,11016] |
| Dosing (1 = 4 times/week, vs. daily) | 1.940*** [1.677,2.244] | 1503 [1176,1999] | 2.135*** [1.807,2.522] | 2048 [1496,3134] |
| Venue (1 = private hospital, vs. government hospital) | 0.905** [0.824,0.994] | -226 [-430,-8] | 0.921 [0.829,1.024] | -221 [-492,77] |
| **Microbicide** | | | | |
| Cost (INR) | 0.969*** [0.965,0.974] | | 0.972*** [0.965,0.978] | |
| Side effects (1 = minor, vs. none) | 0.361*** [0.322,0.405] | -33 [-38,-28] | 0.390*** [0.340,0.446] | -33 [-42,-27] |
| Efficacy (1 = 99%, vs. 50%) | 11.973*** [8.421,17.022] | 79 [69,92] | 8.503*** [5.783,12.504] | 74 [63,90] |
| Dosing (1 = no applicator vs. applicator) | 0.792*** [0.645,0.973] | -7 [-14,-1] | 0.741*** [0.626,0.877] | -10 [-16,-5] |
| Venue (1 = pharmacy vs. CBO) | 0.876* [0.761,1.008] | -4 [-9,0] | 0.870* [0.755,1.001] | -5 [-12,0] |
| **Vaccine** | | | | |
| Cost (INR 000s) | 0.624*** [0.564,0.690] | | 0.702*** [0.614,0.803] | |
| Side effects (1 = minor, vs. none) | 0.170*** [0.110,0.262] | -3758 [-5185,-2667] | 0.329*** [0.240,0.450] | -3150 [-5329,-2004] |
| Efficacy (1 = 99%, vs. 50%) | 11.413*** [7.300,17.844] | 5156 [4397,6106] | 9.593*** [4.771,19.289] | 6398 [4701,9232] |
| Dosing (1 = 1 dose/year vs. 3 doses/year) | 0.572*** [0.501,0.654] | -1182 [-1580,-866] | 0.521*** [0.419,0.648] | -1845 [-3066,-1143] |
| Venue (1 = private hospital, vs. government hospital) | 0.922 [0.788,1.08] | -172 [-535,164] | 0.964 [0.755,1.230] | -105 [-870,630] |

* p<0.1;

** p<0.05;

*** p<0.01

INR, Indian rupee

For vaccines, those engaged in sex work were similarly more price sensitive (OR for cost 0.71 vs. 0.63) than those not engaged in sex work. However, the preference for efficacy was more equivocal, with greater preference among those not engaged in sex work (odds ratio 11.31 vs. 10.16), but greater mWTP among those engaged in sex work (INR 6,663 vs. INR 5,162). This difference in results between preferences and mWTP reflects the conflicting effects of greater preferences and greater price sensitivity among those not engaged in sex work. Those not engaged in sex work also displayed greater preference for avoiding minor side effects (OR 0.19 vs. 0.28), although there was little difference in mWTP, again due to the offsetting effect of greater price sensitivity among participants not engaged in sex work. As with risk behavior, the analysis by sex work status revealed a greater preference for efficacy of

**Table 7. DCE analysis, stratified by sex work.**

| Product/Attribute (Levels) | MSM engaged in sex work | | MSM not engaged in sex work | |
|---|---|---|---|---|
| | Adjusted odds ratio (95% CI) | Marginal willingness-to-pay (95% CI) | Adjusted odds ratio (95% CI) | Marginal willingness-to-pay (95% CI) |
| **PrEP** | | | | |
| Cost (INR 000s) | 0.692*** [0.614,0.781] | | 0.636*** [0.556,0.728] | |
| Side effects (1 = minor, vs. none) | 0.293*** [0.220,0.390] | -3336 [-4305,-2728] | 0.297*** [0.217,0.405] | -2689 [-3368,-2189] |
| Efficacy (1 = 99%, vs. 50%) | 24.287*** [16.334,36.112] | 8675 [6436,12978] | 15.707*** [9.182,26.868] | 6096 [4501,8732] |
| Dosing (1 = 4 times/week, vs. daily) | 1.968*** [1.677,2.311] | 1842 [1355,2673] | 2.112*** [1.82,2.45] | 1655 [1263,2312] |
| Venue (1 = private hospital, vs. government hospital) | 1.000 [0.902,1.109] | 0 [-260,333] | 0.826*** [0.748,0.911] | -424 [-669,-210] |
| **Microbicide** | | | | |
| Cost (INR) | 0.967*** [0.961,0.974] | | 0.972*** [0.967,0.977] | |
| Side effects (1 = minor, vs. none) | 0.349*** [0.307,0.396] | -32 [-38,-27] | 0.393*** [0.344,0.450] | -33 [-41,-27] |
| Efficacy (1 = 99%, vs. 50%) | 16.407*** [10.048,26.788] | 85 [73,99] | 7.707*** [5.557,10.690] | 72 [62,85] |
| Dosing (1 = no applicator vs. applicator) | 0.874 [0.733,1.042] | -4 [-9,2] | 0.700*** [0.563,0.871] | -13 [-21,-5] |
| Venue (1 = pharmacy vs. CBO) | 0.995 [0.837,1.182] | 0 [-6,5] | 0.800*** [0.686,0.933] | -8 [-15,-2] |
| **Vaccine** | | | | |
| Cost (INR 000s) | 0.706*** [0.603,0.826] | | 0.625*** [0.57,0.686] | |
| Side effects (1 = minor, vs. none) | 0.277*** [0.185,0.413] | -3692 [-6825,-2256] | 0.185*** [0.133,0.258] | -3590 [-4686,-2739] |
| Efficacy (1 = 99%, vs. 50%) | 10.164*** [5.755,17.952] | 6663 [5043,10365] | 11.311*** [7.866,16.263] | 5162 [4480,6041] |
| Dosing (1 = 1 dose/year vs. 3 doses/year) | 0.582*** [0.485,0.699] | -1556 [-2883,-931] | 0.535*** [0.468,0.612] | -1331 [-1746,-997] |
| Venue (1 = private hospital, vs. government hospital) | 1.011 [0.831,1.231] | 33 [-575,695] | 0.891* [0.778,1.02] | -246 [-564,49] |

* *p*<0.1;

** *p*<0.05;

*** p<0.01

INR, Indian rupee

microbicides among those engaged in sex work (OR 16.41 vs. 7.71). However, dosing of microbicides, and venue for receiving microbicides, were only statistically significant for those not engaged in sex work, who showed greater preference and mWTP for an applicator and receiving through a CBO rather than a pharmacy.

## Discussion

A key objective of India's National HIV/AIDS Strategic Plan is to reduce new infections by 80% by 2024 [38], which necessitates increasing access to and use of HIV prevention products and promoting combination HIV prevention approaches. In this context, it is crucial to

understand preferences of MSM towards new and emerging HIV prevention products such as PrEP, rectal microbicides and HIV vaccines. Specifically, this paper focused on understanding stated preferences for five product attributes–efficacy, cost, dosing schedule, side-effects and venue. Overall, the results show that while efficacy is the most important attribute, Indian MSM vary in their preferences for HIV prevention product attributes, especially in relation to their levels of education and income, and engagement in sex work and HIV risk behavior. This study contributes to the literature by focusing on MSM in India and quantifying their stated preferences for HIV prevention product attributes, given that most studies addressing willingness to use PrEP, rectal microbicides and HIV vaccines among Indian MSM have used qualitative approaches [7, 30].

Across all three products, four attributes were significant predictors of acceptability—in decreasing order of preference, efficacy, side-effects, dosing schedule, and venue. Delivery venue was a significant predictor of the acceptability of PrEP and rectal microbicides, but not HIV vaccines. Among the significant attributes, efficacy was found to be the most important predictor of acceptability across all three products. Given that the effectiveness of condoms against HIV transmission is estimated at 80–85% among MSM [39], and PrEP, when taken as prescribed, is 99% effective in reducing the risk of contracting HIV [40], participants may expect similarly high levels of efficacy for rectal microbicides and HIV vaccines.

The presence of side-effects, when compared to no side-effects, reduced acceptability across all products. Similar to the present study, other studies have shown that concerns about side-effects may deter people from using new HIV prevention technologies like HIV vaccines [41], rectal microbicides [30] and PrEP [7]. Dosing schedule was a significant predictor of acceptability across all three products; however, this result may be more informative within each product. Unlike efficacy, the levels of dosing schedules differed across products, reflecting expectable product differences (such as longer duration of protection expected from a vaccine) and real-world data on oral PrEP use. Participants' preference for intermittent (or 'on-demand') versus daily PrEP may reflect a perception that it is a more convenient dosing schedule, given concerns about adherence as well as side-effects associated with daily dosing [7]. Studies have shown conflicting results on preferences for intermittent or daily PrEP use [42] as well as the impact of daily life practices on HIV prevention product choices [43]. Dosing preferences also may reflect perceptions about other attributes, such as associations with higher efficacy, side-effects, and cost. For example, some MSM in India associated highly efficacious products with more serious side-effects [31], and may have concerns about costs (product and travel) associated with higher frequency dosing schedules.

Participants' preference for subsidized (lower) price, beyond merely paying less, may be related to their being recruited largely through CBOs providing HIV services, which generally provide free condoms and HIV testing, as well as free STI treatment services in government hospitals.

Subgroup analyses revealed no differences between low- and high-income groups in terms of most product attributes, except that the low-income group preferred government hospitals. This may be explained by the fact that this is their usual source of care given their inability to afford a private hospital. Higher income MSM may exercise the option of paying for private hospital services to avoid discrimination and long wait times in government hospitals [44]. Low-income participants were relatively less sensitive to the price of PrEP than high-income participants, which may reflect their heightened vulnerability to HIV transmission. For both vaccines and PrEP, low-income participants had a greater preference for high efficacy than high-income participants, which also may reflect their higher levels of vulnerability to HIV infection. The reasons for greater preference for high efficacy microbicides among the high-income group are unclear.

Finally, a vaccine is the only product where a significant preference for venue emerged in the comparison, with low-income participants preferring vaccines to be made available through government rather than private hospitals. This may be explained by low-income participants' greater familiarity with government hospital services, particularly as they are often accompanied by peers from CBOs, and the expectation that products distributed from government hospitals will be made available for free, if not at a subsidized price, for people of lower economic status [7]. The DCE results stratified by education were similar to those stratified by income, likely reflecting the association between education and income.

The DCE results stratified on sex work engagement revealed that for both PrEP and vaccines, those engaged in sex work were slightly less price sensitive than those not engaged in sex work; this may reflect PrEP and vaccines being seen as particularly beneficial in the sex work environment. A qualitative study among MSM in India revealed the utility of PrEP as a discreet prevention method that obviated the need for negotiation with partners, some of whom rejected condom use [7]. The present study indicates that participants engaged in sex work had a greater preference for higher efficacy PrEP and rectal microbicides than those not engaged in sex work, possibly reflecting their ongoing context of risk. Differential venue preferences, and particularly the lack of a significant preference for government versus private hospitals among participants engaged in sex work, may reflect concerns about PrEP stigma from peers irrespective of venue, as revealed in a qualitative study with MSM in India [7]. Countervailing influences of anticipated discrimination in government hospitals and concerns about high PrEP prices in private hospitals also might have also resulted in lack of a clear venue preference.

For microbicides, dosing and venue were only statistically significant for those not engaged in sex work, who showed greater preference for an applicator (vs. no applicator) and receiving microbicides through a CBO rather than a pharmacy. Applicator preference may be more of a prerogative of MSM not involved in sex work, as sex workers may experience challenges in always having an applicator on hand and navigating its use before sex [45], with other studies similarly revealing challenges around product portability and assembly [46]. Preferences for a CBO venue may be explained by reports from some MSM in India that pharmacists consider PrEP buyers to be HIV-positive, promiscuous or MSM, with anticipated stigma making pharmacies an unwelcoming venue [7].

## Limitations

Our study has several limitations in addition to its strengths. Oral PrEP has proven effective in clinical trials and demonstration projects around the world, while rectal microbicides and HIV vaccines remain hypothetical products in the development pipeline, which may influence acceptability and preferences. However, PrEP use remains extremely low in India; from the perspective of participants, this likely mitigates what might be seen as a bifurcation between 'real' and hypothetical in the products modeled. It is also an inherent limitation of DCEs that the preferences revealed are contingent on the product attributes and values modeled. The present study's use of categorical variables with two levels of each attribute was based on formative qualitative research and pilot DCEs with the study population, following best practices in the development of the DCEs. Although relevant attributes and levels of products in the development pipeline may change as they near approval, this is commensurate with our aim to prepare for future implementation of HIV prevention products in the development pipeline, a benefit of DCE methods.

The time period of data collection is a further study limitation as willingness to use prevention technologies and preferences may shift over time. Additionally, the study design did not

allow direct comparison of preferences across the products, as we deemed it an undue cognitive burden for participants to indicate choices for three different products each with different attribute profiles. However, we successfully randomized participants to each product, with each attribute depicted with color illustrations to facilitate comprehensibility, and we identified high willingness to use the products. The fact that oral PrEP is still not widely accessible to MSM across India, with the other two products still in the research pipeline, suggests that the findings remain applicable. And while the use of chain-referral sampling expanded recruitment to MSM at risk beyond those who are clients of established CBOs, reliance on physical hotspots and referrals may not reach MSM who use virtual sites (i.e., smartphone-based/online dating apps), identified as having low connection to HIV services [47]. Thus, future modeling of preferences for HIV prevention technologies should include MSM recruited from virtual sites [47].

Finally, we did not model long-acting injectable (LAI-)PrEP, which has been demonstrated to provide greater protection against HIV infection than oral PrEP among MSM, largely attributed to challenges in adherence to daily oral PrEP [48]. However, LAI-PrEP was not yet deemed effective at the time of data collection for the present study. In effect, we modeled important attributes of LAI-PrEP (i.e., an injection administered every two months) as a durable protection method by assessing preferences for a future HIV vaccine with 3-month (vs. 1-year) efficacy. The belated licensure of oral PrEP in India, and the fact that its costs are still not covered by government hospitals, suggests a long road ahead for LAI-PrEP approval. Nevertheless, making PrEP broadly available to MSM and other populations most vulnerable to HIV infection in India remains a critical component of ongoing efforts to achieve UNAIDS targets of no new infections by 2030 [48, 49].

## Conclusions

This study identified key attributes that influence stated choices for three new or emerging biomedical HIV prevention products among MSM in India and delineated heterogeneity in choices across subgroups defined by educational status, income, sex work status and HIV risk behavior. The findings are important for informing research to align candidate product's attributes with end user-preferred attributes; to optimize the effectiveness of new HIV prevention technologies, end users' preferences should be considered throughout product design, testing, and dissemination phases. Moreover, the findings provide potential action points and targets for interventions to support the effectiveness of combination HIV prevention among MSM in India. This includes evidence to support the development of targeted education about product usage and minor side-effects, and identification of venues that are acceptable to end users, as well as highlighting the need to tailor programs and interventions for diverse subgroups in order to optimize uptake of new HIV prevention products among MSM in India.

## Supporting information

**S1 Appendix. Descriptions presented of HIV prevention products.**
(DOCX)

**S1 File. Inclusivity in global research questionnaire.**
(DOCX)

## Author Contributions

**Conceptualization:** Michael P. Cameron, Peter A. Newman, Venkatesan Chakrapani.

**Data curation:** Surachet Roungprakhon.

**Formal analysis:** Michael P. Cameron, Riccardo Scarpa.

**Funding acquisition:** Peter A. Newman, Venkatesan Chakrapani.

**Investigation:** Murali Shunmugam, Dicky Baruah, Ruban Nelson.

**Methodology:** Peter A. Newman, Venkatesan Chakrapani, Riccardo Scarpa.

**Project administration:** Murali Shunmugam, Shruta Rawat, Suchon Tepjan.

**Software:** Riccardo Scarpa.

**Supervision:** Peter A. Newman, Venkatesan Chakrapani.

**Visualization:** Suchon Tepjan.

**Writing – original draft:** Michael P. Cameron, Peter A. Newman, Venkatesan Chakrapani.

**Writing – review & editing:** Michael P. Cameron, Peter A. Newman, Venkatesan Chakrapani, Riccardo Scarpa.

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
