## [Decision Letter · Decision Letter 0]

7 Mar 2024

PONE-D-23-22064Stated Preferences for New HIV Prevention Technologies among Men Who have Sex with Men in India: A Discrete Choice ExperimentPLOS ONE

Dear Dr. Newman,

Thank you for submitting your manuscript to PLOS ONE. After careful consideration, we feel that it has merit but does not fully meet PLOS ONE’s publication criteria as it currently stands. Therefore, we invite you to submit a revised version of the manuscript that addresses the points raised during the review process.

We look forward to receiving your revised manuscript.

Kind regards,

Mario Soliño

Academic Editor

PLOS ONE

Journal Requirements:

3.  “1) Please provide additional information regarding the considerations  made for the participants included in this study. For instance, please discuss whether participants were able to opt out of the study and whether individuals who did not participate receive the same treatment offered to participants.

    2) Please include a complete copy of PLOS’ questionnaire on inclusivity in global research in your revised manuscript. Our policy for research in this area aims to improve transparency in the reporting of research performed outside of researchers’ own country or community. The policy applies to researchers who have travelled to a different country to conduct research, research with Indigenous populations or their lands, and research on cultural artefacts. The questionnaire can also be requested at the journal’s discretion for any other submissions, even if these conditions are not met.  Please find more information on the policy and a link to download a blank copy of the questionnaire here: https://journals.plos.org/plosone/s/best-practices-in-research-reporting. Please upload a completed version of your questionnaire as Supporting Information when you resubmit your manuscript

"I have read the journal’s policy and the authors of this manuscript have the following competing interests: PAN reports serving as an Academic Editor for PLOS ONE. All other authors reported no conflicts of interest."

5. We note that you have indicated that there are restrictions to data sharing for this study. PLOS only allows data to be available upon request if there are legal or ethical restrictions on sharing data publicly. For more information on unacceptable data access restrictions, please see http://journals.plos.org/plosone/s/data-availability#loc-unacceptable-data-access-restrictions.   

Additional Editor Comments:

Two reviewers have revised your manuscript. One fo them has several doubts with the selection of attributes and levels. You may estimate your models using the continuous variables where possible, instead using transformations with dummies. For example, you can icroproate to the model the attribute efficacy as a contonuous variable (50%, 99%) and to consider a quadratic form for this attribute.

Reviewers' comments:

Reviewer's Responses to Questions

**Comments to the Author**

1. Is the manuscript technically sound, and do the data support the conclusions?

Reviewer #1: Yes

Reviewer #2: Yes

2. Has the statistical analysis been performed appropriately and rigorously? 

Reviewer #1: Yes

Reviewer #2: Yes

3. Have the authors made all data underlying the findings in their manuscript fully available?

Reviewer #1: Yes

Reviewer #2: No

4. Is the manuscript presented in an intelligible fashion and written in standard English?

Reviewer #1: Yes

Reviewer #2: Yes

5. Review Comments to the Author

Reviewer #1: Review to PONE-D-23-22064: Stated Preferences for New HIV Prevention Technologies among MSM in India

The paper focuses on the (hypothetical) willingness to use three HIV prevention technologies among men who have sex with men in India. It represents a comprehensive and methodologically sound study with relevant findings, such as the variability of preferences based on sociodemographic and risk behavior factors. However, some aspects could benefit from refinement for enhanced clarity:

Comment 1: The current presentation of the study design is difficult to follow. While it seems that all necessary details regarding recruitment, study design, and the DCE are included, they are not presented in a cohesive and comprehensible manner. For example, one may wonder how the respondents were sampled or how the DCE was designed, only to get an explanation in later sections. This limits the flow of reading considerably. I recommend fundamentally rewriting and reorganizing the sections on study design, sample and data collection, and DCE development to improve clarity and coherence.

Comment 2: The description of the process for assigning individuals to one of the three products is ambiguous and requires further explanation. It remains unclear whether the three products are linked to the three different versions of the questionnaire.

Comment 3: The use of chain-referral sampling raises concerns about the potential for selection bias and survey falsification. I recommend that the authors discuss any quality control measures implemented to ensure that respondents genuinely met the inclusion criteria. To my understanding, there's a risk that participants might falsely have identified as part of the target group to access the study incentives. Addressing this concern helps validating the study's findings as well as strength the paper.

Comment 4: The burden of eight choice scenarios, each including five cards with different attribute level combinations, seems considerable and could lead to participation fatigue. This fatigue might affect the reliability of responses. I suggest that the authors control for effects of survey fatigue, possibly by focusing on a smaller set of decisions made by each respondent, such as only the first 3-4 decisions, as a robustness check.

Comment 5: Ideally, the authors would make their Stata code and the choice scenarios created in Ngene available as part of the supplement so that other researchers can build on their work.

Reviewer #2: This manuscript presents the results of a DCE in India (two Indian cities) to elicit willingness to use a HIV prevention technology, described using different levels of five attributes. Three technologies were analysed, through three different questionnaire versions and samples: 1) HIV pre-exposure prophylaxis (PrEP), (2) rectal microbiocides and (3) HIV vaccines.

Strengths:

The attempt to quantify preference intensity (WTP) about the attributes (characteristics) to obtain relevant information to guide public health decisions and improve their effectiveness.

Weaknesses:

(1) Results about preferences for the three products are not comparable. Results can support only conclusions about the preferences for attributes and levels for each product. (2) There is no literature revision about published articles regarding DCE use in HIV prevention technologies, despite the high amount of literature on the issue (there is a systematic review of 2021 in the Patient-patient centered outcomes research Journal, and in many others like the Journal of International Aids Society or more general like the Journal of Choice Modelling). Methods and results must be compared and discussed taking into account previous literature. (3) Selection of attributes and their levels is not enough explained.

Major issues to be addressed:

1) Improve manuscript to present a review of economic valuation literature regarding DCE and HIV prevention.There is a systematic review of 2021 in the Patient-patient centered outcomes research Journal, and in many others like the Journal of International Aids Society or more general like the Journal of Choice Modelling. Methods and results must be compared and discussed taking into account previous literature.

2) Authors must explain the process followed to select attributes and levels. Levels of some crucial attributes (like efficacy, 50%, 99%; side effects, none, minor) may have conditioned their high significance in the analysis. Also, it would be important to see the information provided before the choice, to check the scenario and previous information, which could be crucial for respondents to fully understand the task.

Recommendation: Major revisions.

6. PLOS authors have the option to publish the peer review history of their article (what does this mean?). If published, this will include your full peer review and any attached files.

Reviewer #1: No

Reviewer #2: No

---

## [Author Response · Author response to Decision Letter 0]

1 Jul 2024

1 JUL 2024

We have now revised the Data Availability Statement to more specifically describe the reasons for not being able to share data publicly (i.e., it contains sensitive patient information and is owned by a third-party organization), the name of the IRB that imposed the restrictions, and an email address to which data access requests may be sent. We trust this addresses all dimensions of PLOS ONE's requirements.

-----

Article ID: PONE-D-23-22064

Title: Stated Preferences for New HIV Prevention Technologies among Men Who have Sex with Men in India: A Discrete Choice Experiment

Journal Requirements:

Response: We have reviewed the manuscript and confirm that it meets PLOS ONE’s style requirements.

3 - 1) Please provide additional information regarding the considerations made for the participants included in this study. For instance, please discuss whether participants were able to opt out of the study and whether individuals who did not participate receive the same treatment offered to participants.

Response: We now provide further details in the manuscript text regarding these considerations at the beginning of the Methods section.

Page 6, paragraph 2:

“Participation was completely voluntary. Individuals who received an invitation to participate could choose whether or not to visit the community organization to do so. Participants who initiated the questionnaire could also opt out of completion with no penalty; however, since all data were anonymized, their responses could not be removed after survey completion, as explained in the informed consent. All individuals had access to HIV prevention services from the partner CBOs whether or not they chose to participate in the study.”

3 - 2) Please include a complete copy of PLOS’ questionnaire on inclusivity in global research in your revised manuscript. Our policy for research in this area aims to improve transparency in the reporting of research performed outside of researchers’ own country or community. The policy applies to researchers who have travelled to a different country to conduct research, research with Indigenous populations or their lands, and research on cultural artefacts. The questionnaire can also be requested at the journal’s discretion for any other submissions, even if these conditions are not met. Please find more information on the policy and a link to download a blank copy of the questionnaire here: https://journals.plos.org/plosone/s/best-practices-in-research-reporting. Please upload a completed version of your questionnaire as Supporting Information when you resubmit your manuscript

Response: As requested, we have uploaded a completed version of the questionnaire on inclusivity in global research.

"I have read the journal’s policy and the authors of this manuscript have the following competing interests: PAN reports serving as an Academic Editor for PLOS ONE. All other authors reported no conflicts of interest."

Response:

As requested, we have included an updated Competing Interests statement in our cover letter, which includes the following text:

“This does not alter our adherence to PLOS ONE policies on sharing data and materials.” 

5. We note that you have indicated that there are restrictions to data sharing for this study. PLOS only allows data to be available upon request if there are legal or ethical restrictions on sharing data publicly. For more information on unacceptable data access restrictions, please see http://journals.plos.org/plosone/s/data-availability#loc-unacceptable-data-access-restrictions. 

Response: Data are owned by the Humsafar Trust, and the Centre for Sexuality and Health Research and Policy, in India. Due to the sensitivity of participant information about sexuality and HIV in India, the Institutional Review Board (IRB) of the Humsafar Trust has imposed restrictions on public sharing of the data sets. Requests for access to de-identified data sets for the results presented will be granted on a case-by-case basis by contacting the IRB Coordinator of the Humsafar Trust, Mr. Sandeep Mane, at info@humsafar.org or +91-9892940966. 

Additional Editor Comments: 

Two reviewers have revised your manuscript. One of them has several doubts with the selection of attributes and levels. You may estimate your models using the continuous variables where possible, instead using transformations with dummies. For example, you can icroproate to the model the attribute efficacy as a continuous variable (50%, 99%) and to consider a quadratic form for this attribute.

Response: We appreciate this concern, however all attributes in the DCE took on only two values. So, while estimating the models using continuous rather than categorical variables is possible, the re-estimation would be functionally identical. For example, efficacy took on two levels (50% and 99%). If you divide the coefficient estimate from the current model estimations by 49, then you obtain the marginal effect for a 1-percentage-point increase in efficacy. Also, because efficacy takes on only two values, a model with a quadratic function in efficacy cannot be estimated as the efficacy and efficacy^2 variables will be collinear. Thus we have not changed this in the text; however, in deference to Reviewer 2 we now indicate the use of categorical variables with only 2 levels as a potential study limitation, in addition to potential changes in the relevant attributes of products still in the development pipeline. We also have added substantial details about the process of selecting attributes and levels, as in response to Reviewer 2, comment 2.

Page 2, paragraph 3:

“It is also an inherent limitation of DCEs that the preferences revealed are contingent on the product attributes and values modeled. The present study’s use of categorical variables with two levels of each attribute was based on formative qualitative research and pilot DCEs with the study population, following best practices in the development of the DCEs. Although relevant attributes and levels of products in the development pipeline may change as they near approval, this is commensurate with our aim to prepare for future implementation of HIV prevention products in the development pipeline, a benefit of DCE methods.”

Page 9, paragraph 3 – page 10, paragraph 4:

“We used a multistage process to develop the DCEs, in accordance with best practice guidelines [12]. First, we conducted a literature review and formative qualitative research on each prevention technology with the focal study population [7,30,31]. Based on this information, we identified the most salient attributes for these HIV prevention technologies: concerns about the degree of efficacy, dosing frequency, and side effects emerged across our qualitative research on hypothetical HIV vaccines, microbicides, and PrEP [7,30,31]. Additionally, MSM expressed concerns about stigma if it were disclosed that they were accessing these HIV prevention products, such as being judged as sexually “promiscuous” or being “outed” to their families as MSM. This invoked the potential importance of the venue for accessing HIV prevention products as a determinant of product acceptability and use.

Second, we conducted a pilot DCE study with MSM (n = 16) recruited from CBO partners in India; this provided a priori guidance on the model coefficients and guided the selection of attributes (e.g., efficacy) and levels (e.g., 50% vs 99%) for each product in the DCE [32]. It also supported the feasibility of implementing DCEs with this population.

Third, we incorporated methodological recommendations for choice elicitation tasks [32]: alternative attribute levels (e.g., 50% vs 99% efficacy; none vs. minor side effects) need to be sufficiently distinct to be comprehensible to participants, most of whom are unlikely to be able to discern the difference between 50% and 65% efficacy, for example, or a 15% vs. 25% chance of fever or headaches. Our recruitment aimed to reach vulnerable populations of largely low socioeconomic status MSM, many without college education and with limited economic opportunities, a substantial proportion of whom rely on sex work for income. This underscored the importance of selecting attribute levels to facilitate comprehension among participants with low numeracy. 

Finally, the very high efficacy of oral PrEP in clinical trials informed our use of 99% as a level. As a result, 99% efficacy was used as the upper value for all 3 products, in contrast with 50% to clearly signify partial efficacy.”

REVIEWER #1

The paper focuses on the (hypothetical) willingness to use three HIV prevention technologies among men who have sex with men in India. It represents a comprehensive and methodologically sound study with relevant findings, such as the variability of preferences based on sociodemographic and risk behavior factors. However, some aspects could benefit from refinement for enhanced clarity:

Comment 1: The current presentation of the study design is difficult to follow. While it seems that all necessary details regarding recruitment, study design, and the DCE are included, they are not presented in a cohesive and comprehensible manner. For example, one may wonder how the respondents were sampled or how the DCE was designed, only to get an explanation in later sections. This limits the flow of reading considerably. I recommend fundamentally rewriting and reorganizing the sections on study design, sample and data collection, and DCE development to improve clarity and coherence.

Response: We have now substantially rewritten and reorganized the Methods section, which has improved the flow of reading. In accordance with the reviewer’s concern, we first describe the overall Study Design and Context, then include a separate section on Sampling and Recruitment, and thereafter we provide details about the DCE Experimental Design and DCE Development. This helps to clarify the new section on Data Collection, and then Measures, which come afterward. This reorganization also provides additional descriptions to address the Reviewer’s comments 2, 3 and 4 below.

Comment 2: The description of the process for assigning individuals to one of the three products is ambiguous and requires further explanation. It remains unclear whether the three products are linked to the three different versions of the questionnaire.

Response: We now provide a clearer description of the assignment of individuals to one of three survey versions, each with a DCE of one product. The three survey versions were the same except for the target product of the DCE.

Page 11, paragraph 3 – page 12, paragraph 1:

“To mitigate potential respondent fatigue, we created three versions of the questionnaire, each of which was identical except for the DCE. Participants were randomly assigned to one of the three versions including a DCE for one prevention technology. The DCE experimental design (based on Bayesian D-error minimization) contained 32 choice scenarios of five hypothetical product alternatives for each of the three products. To reduce cognitive burden on participants, these were each blocked in four groups of eight choice scenarios. Random allocation to survey versions and blocking within each product DCE were programmed on the Tablet devices.”

Comment 3: The use of chain-referral sampling raises concerns about the potential for selection bias and survey falsification. I recommend that the authors discuss any quality control measures implemented to ensure that respondents genuinely met the inclusion criteria. To my understanding, there's a risk that participants might falsely have identified as part of the target group to access the study incentives. Addressing this concern helps validating the study's findings as well as strength the paper.

Response: As the reviewer indicates, there are risks in chain-referral sampling, including misrepresentation of one’s eligibility and duplicate participation. We now describe several quality control measures that we implemented to address these concerns. Benefits of the chain-referral method are that it enables one to reach potential participants beyond CBO clientele—often those recruited for HIV research among community samples—who may be at greater risk for HIV due to lack of engagement with HIV education and prevention services. As we now describe in the manuscript

Page 8, paragraph 2:

“We employed several quality control measures to mitigate risks in chain-referral sampling; for example, individuals might present at the study site, having not been referred, and may misrepresent their sexual identity or behavior to enable them to receive study honoraria. For one, peer recruiters were employed by existing CBOs and well-established in the MSM community and recruitment sites; and they were trained to assess the eligibility of potential participants. Second, each recruiter maintained a log register that tracked individuals referred by seeds—directly invited by the peer recruiter—and then cross-checked whether the same individuals were also referred by other seeds (based on duplication in names, ages, and phone numbers in the logs). Third, pre-numbered coupons were issued to all participants, each with a unique identification number (UID), to invite others in

---

## [Editor Report · Decision Letter 1]

10 Jul 2024

Stated Preferences for New HIV Prevention Technologies among Men Who have Sex with Men in India: A Discrete Choice Experiment

PONE-D-23-22064R1

Dear Dr. Newman,

We’re pleased to inform you that your manuscript has been judged scientifically suitable for publication and will be formally accepted for publication once it meets all outstanding technical requirements.

Kind regards,

Mario Soliño

Academic Editor

PLOS ONE
---

## [Editor Report · Acceptance letter]

15 Jul 2024

PONE-D-23-22064R1 

PLOS ONE

Dear Dr. Newman, 

I'm pleased to inform you that your manuscript has been deemed suitable for publication in PLOS ONE. Congratulations! Your manuscript is now being handed over to our production team.

Kind regards, 

on behalf of

Dr. Mario Soliño 

Academic Editor

PLOS ONE